

# Geostationary aerosol retrievals of extreme biomass burning plumes during the 2019-20 Australian bushfires

Daniel J. V. Robbins[1,2], Caroline A. Poulsen[1,3], Steven T. Siems[1,2,4], Simon R. Proud[5,6], Andrew T. Prata[1,4], Roy G. Grainger[7], and Adam C. Povey[7,†]

[1]School of Earth, Atmosphere and Environment, Monash University, Clayton, Victoria 3800, Australia
[2]ARC Centre of Excellence for Climate Extremes, Monash University, Melbourne, VIC 3800, Australia
[3]Science and Innovation Group, Australian Bureau of Meteorology, Melbourne, VIC 3001, Australia
[4]ARC SRI Securing Antarctica's Environmental Future, Melbourne, VIC 3800, Australia
[5]STFC RAL Space and the National Centre for Earth Observation, Rutherford Appleton Laboratory, Didcot, OX11 0QX, UK
[6]National Centre for Earth Observation, Space Park Leicester, Leicester, LE4 5SP, UK
[7]National Centre for Earth Observation, University of Oxford, Clarendon Laboratory, Parks Road, Oxford OX1 3PU, UK
[†]Now at School of Physics and Astronomy, University of Leicester, University Road, Leicester, LE1 7RH, UK

**Correspondence:** Daniel Robbins (daniel.robbins@monash.edu)

**Abstract.**

Extreme biomass burning (BB) events, such as those seen during the 2019-20 Australian bushfire season, are becoming more frequent and intense with climate change. Ground-based observations of these events can provide useful information on the macro- and micro-physical properties of the plumes, but these observations are sparse, especially in regions which are at risk of

intense bushfire events. Satellite observations of extreme BB events provide a unique perspective, with the newest generation of geostationary imagers, such as the Advanced Himawari Imager (AHI), observing entire continents at moderate spatial and high temporal resolution. However, current passive satellite retrieval methods struggle to capture the high values of aerosol optical thickness (AOT) seen during these BB events. Accurate retrievals are necessary for global and regional studies of shortwave radiation, air quality modelling and numerical weather prediction. To address these issues, the Optimal Retrieval of Aerosol

and Cloud (ORAC) algorithm has used AHI data to measure extreme BB plumes from the 2019-20 Australian bushfire season. The sensitivity of the retrieval to the assumed optical properties of BB plumes is explored by comparing retrieved AOT with AERONET L1.5 data over the AERONET site at Tumbarumba, New South Wales, between 1 December 2019 00:00 UTC to 3 January 2020 00:00 UTC. The study shows that for AOT values > 2, the sensitivity to assumed optical properties is substantial. The ORAC retrievals and AERONET data are compared against the JAXA Aerosol Retrieval Product (ARP), MODIS Deep

Blue over land, MODIS MAIAC SLSTR SYN and VIIRS Deep Blue products. The comparison shows the ORAC retrieval significantly improves coverage of optically thick plumes relative to the JAXA ARP, with approximately twice as many pixels retrieved and peak retrieved AOT values 1.4 higher than the JAXA ARP. The ORAC retrievals have accuracy scores between 0.742 - 0.744 compared to the values of 0.718 - 0.833 for the polar-orbiting satellite products, despite successfully retrieving approximately 28 times as many pixels over the study period as the most successful polar-orbiting satellite product. The AHI

and MODIS satellite products are compared for three case studies covering a range of BB plumes over Australia. The results show good agreement between all products for plumes with AOT values ≤ 2. For extreme BB plumes, the ORAC retrieval



finds values of AOT > 15, significantly higher than those seen in events classified as extreme by previous studies although with high uncertainty. A combination of hard limits in the retrieval algorithms and misclassification of BB plumes as cloud prevent the JAXA and MODIS products from returning AOT values significantly greater than 5.

## 1 Introduction

Aerosols are an important component of the Earth's atmosphere with significant effects on the Earth's radiation budget (Bellouin et al., 2020). Overall, aerosols are considered to have a net cooling effect, but there remains a high level of uncertainty regarding the effects of some aerosol species, such as organic and black carbon (Arias et al., 2021). Aerosols affect cloud formation and weather. For example Connolly et al. (2012) found that the inclusion of aerosols in modelling of the convective system known as *Hector* in Northern Australia improved the accuracy of the storm's development. Huang and Ding (2021) demonstrated that aerosols can have a significant impact on temperature forecasts. Biomass burning (BB) events release large amounts of black and brown carbon aerosol into the atmosphere (Andreae, 2019; Pan et al., 2020), which can directly affect the local and global radiation budget by absorbing and scattering short-wave radiation, known as the "direct radiation effect", or through interaction with clouds, known as the "semi-direct effect" and "indirect effect" (Ackerman et al., 2000; Haywood and Boucher, 2000; Bellouin et al., 2005; Lisok et al., 2018; Matus et al., 2019; Shi et al., 2019; Liu et al., 2020). Characterisation of aerosols is important for numerical weather prediction (NWP), where assimilation of aerosols is needed to ensure accurate incident radiation and temperature predictions (Huang and Ding, 2021; Juliano et al., 2022), as well as in global reanalysis products of aerosols (Flemming et al., 2017; Gelaro et al., 2017). BB aerosols affect air quality, most notably through the emission of particulate matter under 2.5 μm in diameter (PM2.5), which is know to have serious impacts on people's health and can remain suspended in air for several days (Engel-Cox et al., 2004; Reisen et al., 2013; Chen et al., 2017; Sorek-Hamer et al., 2020).

### 1.1 Measuring Aerosols

Observations of aerosols, such as BB, are important for evaluating their uncertainties in models and impacts (Schutgens et al., 2020; Vogel et al., 2022; Zhong et al., 2022). Ground-based instrumentation, such as the AErosol RObotic NETwork (AERONET) (Holben et al., 1998), can accurately monitor aerosol properties in the atmosphere directly above the site location and are a useful tool for building global aerosol climatologies. AERONET data is used in the development of aerosol classifications and models for satellite retrieval products (Omar et al., 2009; Lyapustin et al., 2018). However, AERONET sites are sparse, with very few sites in regions such as Australia.

Satellites provide global observations. Active instruments in low Earth orbit (LEO), such as the Cloud-Aerosol Lidar with Orthogonal Polarization (CALIOP) (Winker et al., 2009), retrieve profiles of aerosols in the atmosphere along the sub-orbital path, but have a narrow swath, greatly limiting coverage. Passive LEO instruments, such as the Moderate Resolution Imaging Spectroradiometer (MODIS) (Justice et al., 2002) and Sea and Land Surface Temperature Radiometer (SLSTR) (Coppo et al., 2010), have wider swaths and can retrieve aerosol properties over a larger area with moderately high resolution and





shorter revisit times. Passive geostationary (GEO) satellite instruments, such as the Advanced Himawari Imager (AHI) on-
board Himawari-8/9 (Bessho et al., 2016), Advanced Baseline Imager (ABI) on-board the GOES-16 and GOES-17 satellites
(Schmit et al., 2017; Goodman, 2020) and the Flexible Combined Imager (FCI) on-board the recently launched MTG-I1 satel-
lite (Holmlund et al., 2021) have been used to provide high temporal and moderately high spatial resolution aerosol products.
Passive satellite instruments rely on retrieval algorithms that require a representation of the surface reflectance and aerosol
microphysical properties alongside cloud masking, all of which are sources of error.

## 1.2 Wildfires in Australia

Biomass burning events have been frequent throughout Australia's history. Indigenous people across Australia engaged in
cultural or cool burning before colonisation and, more recently, states across Australia have moved to prescribed burns as
part of fire management strategies (Gott, 2005; Morgan et al., 2020). There have been several significant uncontrolled BB
events, known as bushfires in Australia, in recent years. such as the Black Saturday bushfires in 2009 and the 2019-20 Black
Summer bushfires, which were exacerbated by hot, dry conditions and a build-up in combustible material (Cruz et al., 2012;
van Oldenborgh et al., 2021). The conditions for fire events are expected to become more intense and extreme with climate
change (Abram et al., 2021; Canadell et al., 2021).

In the case of the 2019-20 Australian bushfire season, over 110 Tg of biomass was burnt, releasing approximately 1.7 Tg
of particulate matter with a diameter < 2.5 µm (PM2.5) into the atmosphere (Li et al., 2021a). The fires had a significant
impact on people's health (Walter et al., 2020; Graham et al., 2021) and directly resulted in the deaths of 34 people (Wen et al.,
2022). It is estimated that there were approximately 417 excess deaths associated with the very high concentrations of PM2.5
(Arriagada et al., 2020) and approximately 3 billion animals were estimated to have either been killed or displaced as a result of
fire (Dickman, 2021). Large fires have also occurred in North America, Siberia and Southern Europe (Markowicz et al., 2016;
Zhuravleva et al., 2017; Ribeiro et al., 2020; Fernández-García et al., 2022), but have not matched the scale of the Australian
bushfires (Boer et al., 2020; Collins et al., 2021). When fires on these scales occur in mountainous regions with large valleys,
Kochanski et al. (2019) demonstrated that smoke can cause inversions and lead to high optical thickness throughout the region.
This suggests that, in regions of Australia with similar topography, such as the SE coast of Australia which is dominated by the
Great Dividing Range (see Fig. 1), BB events can lead to high optical thicknesses and high levels of pollution. These optically
thick plumes are clearly visible in satellite imagery, as well as in the very poor air quality measurements seen along the SE
coast of Australia during the 2019-20 bushfire period (Graham et al., 2021). This paper will focus on the 2019-20 fires as a
study of high impact, high aerosol loading events.

## 1.3 Satellite Measurements

Previous analysis of the 2019-20 fires has mostly focused on optically thin stratospheric layers or publicly available tropo-
spheric satellite aerosol optical thickness (AOT) products (Torres et al., 2020; Chang et al., 2021; Hirsch and Koren, 2021;
Li et al., 2021b; Attiya and Jones, 2022; Papanikolaou et al., 2022; Sellitto et al., 2022; Isaza et al., 2023). Li et al. (2021b)
found that $AOT_{440\,nm}$ values over the Tumbarumba AERONET site reached 2.74. Torres et al. (2020) found TROPOspheric



Monitoring Instrument (TROPOMI) $AOT_{388nm}$ values > 5 in plumes transported over New Zealand, whereas Papanikolaou et al. (2022) found a maximum in CALIOP AOT of 0.54 in lower tropospheric layers near fire spots. For the summer months of 2019-20 in Victoria state, Chang et al. (2021) found AOT values slightly greater than 0.3 in MODIS L3 monthly data. Upon

closer inspection of the AERONET data used by Li et al. (2021b), $AOT_{500\,nm}$ values > 6 can be seen at the Tumbarumba site (https://aeronet.gsfc.nasa.gov/cgi-bin/data_display_aod_v3?site=Tumbarumba; last access 06 June 2023), which is located at approximately 35.7083° S, 147.9499° E, near Canberra (see Fig. 1), suggesting that these studies have not explored the more optically thick events during the 2019-20 bushfire season.

     A review of passive retrievals for extreme smoke plumes from fires in North America and Siberia offers some insight into

why the retrievals may be restricted. The 2012 Western Siberian fires were found by Zhuravleva et al. (2017) to reach $AOT_{550\,nm}$ of approximately 3.54 with subsequent significant cooling effects on the surface. Markowicz et al. (2016) found that fires in Canada from July 2013 produced $AOT_{500\,nm}$ anomalies up to 1.5 times the climatology for July at several AERONET sites, with similar enhancements observed over Polish and Central European AERONET sites. However, Markowicz et al. (2016) observed that AOT data from SEVIRI and MODIS did not reach AOT values greater than 0.6. Petrenko et al. (2017) showed

that average MODIS BB AOT observations and adjusted GOCART model BB AOT values were less than 2 for regional results of reanalysis-filled MODIS Dark Target observations of almost 900 fires across the globe. These values are all considered extreme, but are significantly less than the peak in $AOT_{500\,nm}$ seen at the Tumbarumba AERONET site during the 2019-20 bushfire season. Eck et al. (2019) demonstrate that the AERONET products are capable of dealing with extrapolated $AOT_{550\,nm}$ values as high as 13, well beyond the upper limits seen in these previous studies. Much of this under prediction is the result

of misclassification of aerosol as cloud and limitations in the retrieval models. For example, the algorithm theoretical basis document (ATBD) for MODIS Multi-Angle Implementation of Atmospheric Correction (MAIAC) algorithm states that the $AOT_{470nm}$ retrieval is limited to a maximum of 4 (Lyapustin and Wang, 2008), well below the peak seen in the Tumbarumba data. Previous work has demonstrated that operational cloud masks for the Himawari-8 AHI instrument misclassify the more extreme smoke plumes as cloud (Robbins et al., 2022), which is also seen in other retrievals of high AOT events (Wei et al.,

2019; Ye et al., 2022). These all prevent proper classification of the values of AOT seen during the 2019-20 Australian bushfire season and limited our ability to study the impacts of this event. In addition, current aerosol retrieval algorithms assume that an aerosol layer is translucent and well mixed, meaning that extreme AOT values are not physically sensible in these types of retrieval.

     Some attempts to overcome these limitations have been made. For example, Mukai et al. (2021) demonstrated that existing

satellite data can be used to carry out retrievals of extreme biomass burning plumes by utilising the UV band of the second-generation global imager to correctly identify optically thick BB plumes and van Donkelaar et al. (2011) showed that simply relaxing the cloud screening of the MODIS aerosol products significantly improved coverage of the biomass burning plumes from the 2010 Moscow fires, but these methods have not been applied to geostationary meteorological satellites.

     In this study, we present a novel method for retrieving BB aerosol optical properties from Himawari-8 AHI using the

Optimal Retrieval of Aerosol and Cloud (ORAC) algorithm (Thomas et al., 2009; Poulsen et al., 2012; McGarragh et al., 2018). Using data from the 2019-20 Australian bushfires, we demonstrate that this retrieval method, in combination with




the cloud mask developed in Robbins et al. (2022), is capable of retrieving extreme values of AOT well beyond what current retrieval algorithms can retrieve. These results demonstrate the limitations of current satellite retrieval products, suggesting that AOTs from the 2019-20 Australian bushfires and other large-scale fires around the globe are likely to have been systemically underestimated, which will alter estimates of the extent of BB plumes and the fires' radiative impact.

## 2 Data Sources

### 2.1 AERONET

The AERONET (Holben et al., 1998; Giles et al., 2019) project is a network of sun-sky photometers located across the globe collecting information on aerosol optical properties, such as the complex refractive index, AOT and classification of aerosols, e.g. fine or coarse. This data is extracted from the raw output using the V3 inversion algorithm developed by Sinyuk et al. (2020), which provides improved cloud clearing over the previous algorithm, as well as being more capable of handling extreme AOT values. The maximum value of AOT reported by AERONET is approximately 7 as the instrument is unable to automatically locate the sun under such large attentulations.

For the period of this study (1st December 2019 - 2nd January 2020), only version 3 level 1.5 AOT data from the Tumbarumba AERONET site was available at the time of publication. The level 1.5 data is cloud-cleared, but has not gone through the full QA screening required to report the data at level 2.0. Level 1.5 data is considered suitable for this work and has been used in similar studies (González et al., 2020; Li et al., 2021b; Yang et al., 2021).

### 2.2 Himawari-8 AHI

The Himawari-8 satellite is a geostationary meteorological satellite operated by the Japanese Meteorological Agency (JMA), which is positioned above 140.7° E and carries AHI. AHI is a passive satellite instrument which produces whole visible disk images every 10 minutes across 16 bands at 0.5-2 km spatial resolution at nadir (see Table 1), with the exception of scheduled "house–keeping times" at 02:40 and 14:40 UTC (Bessho et al., 2016). The raw digital counts for each band, along with the calibration information, is published as L1b Himawari Standard Data (HSD) files.

Two Himawari aerosol products are considered in this study. The Japan Aerospace Exploration Agency (JAXA) produces a level 2 (L2) Aerosol Retrieval Product (ARP) for AHI, which provides aerosol optical thickness (AOT), AOT uncertainty and Angstrom exponent on a regular 0.05° latitude-longitude grid every 10 minutes (Yoshida et al., 2018). The ORAC algorithm is applied to AHI to retrieve AOT and effective radius at the native resolution of AHI (2 km × 2 km at nadir). The calibration outlined in block header 5 of HSD files is used to extract the radiances.

### 2.3 MODIS

MODIS is a passive instrument on-board NASA's Terra and Aqua satellites. The satellites operate in sun-synchronous LEO with the constellation providing rapid return-to-target times, allowing for moderate spatial (0.25-1 km) and daily temporal



coverage. The sun-synchronous orbit means that areas are viewed at approximately the same local solar time by each satellite. This is late morning for Terra and mid afternoon for Aqua.

In this study, AOT values from the Collection 6.1 Deep Blue (DB) (Sayer et al., 2019) and MAIAC (Lyapustin et al., 2018)
algorithms are employed. The DB product (MOD04_L2 for Terra and MYD_L2 for Aqua) is provided at 10 km resolution for each MODIS granule, whereas the MAIAC product (MCD19A2) is provided at 1 km resolution, with data from each MODIS granule regridded to a sinusoidal grid.

## 2.4  SLSTR

The Sea and Land Surface Temperature Radiometer (SLSTR) is a dual-view passive instrument carried on-board the Sentinel-
3A and Sentinel-3B LEO satellites (Coppo et al., 2010), along with the Ocean and Land Colour Instrument (OLCI). For the period of this study, the L2 SYN product is available to compare with AERONET and other retrievals. This product only provides AOT data over land and is regridded to the OLCI native grid (North and Henkel, 2010; Henocq et al., 2018).

## 2.5  VIIRS

The Visible Infrared Imaging Radiometer Suite (VIIRS) on-board NASA's Suomi National Polar-orbiting Partnership (SNPP),
NOAA-20 and NOAA-21 satellites is a passive instrument that provides aerosol optical thickness data at 6 km spatial resolution for each VIIRS granule. Level 2 data is produced using the Deep Blue algorithm over land (Sayer et al., 2018; Hsu et al., 2019). For this study, the AERDB_L2_VIIRS_SNPP collection 1.1 products available over the full validation period are used.

## 3  Methodology

Two types of analysis have been carried out as part of this study:

– A sensitivity study to assumed optical properties in retrieving AOTs from biomass burning plumes in AHI data.

– Case-studies comparing retrieved properties for relatively low AOT plumes over land, as well as for an extreme plume at the coast of SE Australia.

### 3.1  ORAC, Optimal Estimation and State Vectors

Throughout this work, the ORAC algorithm is applied to AHI data to retrieve AOTs at 550 nm, effective radii and (where
possible) heights for all pixels. ORAC is an optimal estimation algorithm that utilises Bayes theorem to minimise a cost function whilst accounting for uncertainties on *a priori* information and satellite measurements (Rodgers, 2000). ORAC contains two approaches to retrieving properties; one for aerosol (Thomas et al., 2009) and one for cloud (Poulsen et al., 2012; McGarragh et al., 2018). The aerosol approach assumes that the aerosol plume is optically thin, well-mixed in the atmosphere and the surface reflectance makes a significant contribution to the top of atmosphere reflectance. Therefore, both AOT and the surface
reflectance are included in the retrieved variables (Thomas et al., 2009). In the case of high AOT BB plumes, like those from the



2019-20 bushfires, the aerosol is thick enough that the surface contributions are a sufficiently minor source of error that they can be held constant during the retrieval. The cloud approach uses an estimate of surface reflectance based on MODIS observations over land and the Cox-Munk algorithm over ocean (Cox and Munk, 1954; Sayer et al., 2010). This approach is used for high AOT plumes and the conventional aerosol approach would be the correct method for dealing with more conventional BB plumes. The novelty of using the cloud approach is that significantly higher values of optical thickness can be retrieved compared to the aerosol approach. The cloud approach uses two radiative transfer models that are considered separately, but resolved simultaneously: the solar radiation component and the thermal radiation component. A full description of this technique can be found in Prata et al. (2022), McGarragh et al. (2018) and Poulsen et al. (2012). Similarly to the methodology described in Prata et al. (2022), cloud fraction is ignored in the state vector as all pixels are assumed to contain some level of aerosol. The cloud mask developed in Robbins et al. (2022) is used to remove cloud pixels that may be erroneously retrieved and contaminate further analysis in case-studies and height comparisons. In addition, a cost threshold is applied, such that pixels which have not adequately fitted the observations are not included in the study.

The ORAC algorithm can evaluate any combination of visible/infrared observations in atmospheric windows (e.g. that lack substantial absorption). In this study, a short-wave (SW) channel retrieval scheme is chosen that uses AHI bands 1 and 3-5, with the 0.51 μm band omitted as it does not match with the corresponding band in MODIS (see Appendix A). The 2.3 μm band is not used as it conflicts with information regarding effective radius that is captured by the 1.6 μm band, as well as being less sensitive to AOT from BB, as AOT values from BB drop off rapidly with wavelength. Smoke, including biomass burning plumes, are usually considered transparent in the long-wave (LW) and literature regarding the optical properties of biomass burning material in the LW is sparse. However, the characterisation of biomass burning by Sutherland and Khanna (1991) reveals that smoke has weak absorption in the AHI 10.4 μm band (see Fig. 2), suggesting that it is possible to retrieve aerosol information from optically thick plumes.

Throughout this study, it is assumed that the smoke particles are spherical. The review of Tumbarumba AERONET data from December 2019 in Li et al. (2021b) indicates that fine-mode AOT dominates, with BB median radius of 0.14-0.25 μm. Therefore, all ORAC retrievals use values of $0.195 \pm 0.055$ μm for the first guess and *a-priori* of effective radius.

## 3.2 Development and Utilisation of Look-Up Tables

The ORAC algorithm relies on offline parameterisation of the multiple scattering properties of a cloud or aerosol layer. This information is stored in look-up tables (LUTs), which are used in the forward model to describe the macro- and micro-physical properties of a cloud or aerosol layer, such as biomass burning plumes. These LUTs are generated using the DIscrete Ordinates Radiative Transfer (DISORT) code (Stamnes et al., 1988). A full description of this generation and the limitations of this method can be found in McGarragh et al. (2018). For this study, 3 new LUTs have been developed for AHI. These are the mean biomass, increased biomass and decreased biomass LUTs. The SW optical properties are derived using data from the Tumbarumba AERONET site during 2019, with the mean values for refractive index (RI) used for mean biomass, the mean RI plus one standard deviation for increased biomass and the mean RI minus one standard deviation for decreased biomass. It should be noted that, although all these LUTs are derived from within the bounds of measurement uncertainty, the decreased





biomass and increased biomass development is an over-simplification and assumes that changes in optical properties have no influence on other properties of the aerosol, which is not true. Nakajima-King plots (Nakajima and King, 1990) for these LUTs (see Fig. 3) indicate that for the effective radii that are observed during the 2019-20 bushfire period, there may be a relatively flat cost surface that leads to a wide range of solutions within the bounds of instrument and model error, leading to relatively high uncertainties at high AOT values. The thermal infra-red (TIR) response is derived from properties described in Sutherland

and Khanna (1991). The 3 versions of the LUTs are used to investigate how sensitive single-view aerosol retrievals of high AOT biomass burning events are to the refractive index

### 3.3 Collocation Methodology

For the sensitivity study, AERONET data between 01 December 2019 UTC 00:00 (inclusive) and 03 January 2020 UTC 00:00 (exclusive) has been collocated with ORAC retrievals, JAXA ARP, MODIS Deep Blue, MODIS MAIAC and SLSTR SYN

data. Beyond January 2020, AERONET data at Tumbarumba may have become contaminated (ash possibly in the collimator and on the lens; Ian Lau, personal communication; 25 November 2021) and other particulates after large fires around this site (González et al., 2020). The period chosen overlaps with the time identified by Li et al. (2021b) at which there were smoke plumes over the Tumbarumba site.

AERONET data is provided between 440 nm to 1640 nm, but shorter wavelengths tend to become saturated at lower values

of optical depth than bands at longer wavelengths (Giles et al., 2019). Eck et al. (2019) demonstrated that during extreme BB events, higher values of AOT are retained for the 675 nm band and extrapolation of AOT at 550 nm from these values allows high AOT events to be studied with AERONET data. Therefore, to ensure that all values of AOT are compared at the same wavelength and reduce the likelihood that data is missed due to saturation of the channel under extreme smoke plumes, the AOT at 675 nm has been extrapolated to 550 nm using the equation,

$$\tau_\lambda = \tau_{\lambda_0} \left( \frac{\lambda}{\lambda_0} \right)^{-\alpha}, \tag{1}$$

where $\tau_\lambda$ is the AOT at some wavelength, $\lambda$, $\tau_{\lambda_0}$ is at some reference wavelength, $\lambda_0$, and $\alpha$ is the Angstrom exponent for that range.

To ensure the best match in time between all products, AERONET data has been averaged in 10 minute segments to match the temporal resolution of AHI. The AERONET data has been binned from the start of an AHI scene time, e.g. 00:00 UTC,

to the approximate end of the scene time, e.g. 00:10 UTC. This introduces some temporal error to the AERONET data, as instantaneous AOTs may be significantly higher or lower than the derived mean. Given the relatively short time period over which this downsampling occurs, it is not expected that there will be significant jumps in AOT and the standard deviations of the temporally downsampled bins are retained within this analysis.

As the JAXA ARP has a spatial resolution of approximately 5 km, whilst ORAC retrieves values at 2 km resolution (at

nadir), two datasets are presented for ORAC retrievals. The first is derived from the collocated pixel that corresponds to the Tumbarumba site, which gives the best match for comparing against AERONET data and is referred to as the pixel-wise



comparison. The second set is for a 3 × 3 pixel region around the site, such that the retrieved values are at approximately 6 km spatial resolution and is referred to as 'spatially-downsampled'. This gives a more suitable comparison to JAXA ARP, but at the expense of accuracy when comparing to AERONET. JAXA ARP data is taken from the pixel collocated with Tumbarumba.

The AOT at 500 nm is extrapolated to 550 nm by using the Angstrom exponent provided for each ARP pixel using Eq. 1.

Several AOT products from polar orbiters are compared in this study. The Deep Blue and MAIAC products from MODIS are included when a MODIS granule is coincident with the Tumbarumba site. As the MAIAC product is provided at 1 km resolution, a 5 × 5 pixel mean about the Tumbarumba site is calculated, with the population mean and standard deviation retained for analysis. Similarly, the SLSTR SYN product is included where available. As the product is provided on the OLCI

grid at 300 m resolution, a 17 × 17 pixel grid about the Tumbarumba site is calculated when a SLSTR overpass is coincident. VIIRS Deep Blue products, at 6 km spatial resolution, are also included where coincident with the Tumbarumba site.

It should be noted that the comparison makes no assumption regarding the heights of the plumes, and have not been parallax corrected. This will introduce some unquantified error into the study. The heights of extreme BB plumes in this region are generally < 3 km, with AHI viewing zenith angles of approximately 40°. This suggests the plumes are shifted by < 2.5 km due

to parallax, which is within the spatial window of the downsampled AHI data. In addition, for all downsampled products, if there is a failed retrieval within the downsampled region, i.e. a pixel is masked such that only the fill value is available, that whole region is flagged as a failed retrieval to ensure that the comparison between the high resolution and lower resolution products is as consistent as possible. To evaluate the skill of flagging an aerosol event, several metrics are presented for this comparison, including the true positive rate (TPR), false positive rate (FPR) and the Kuiper skill score (KSS). The KSS is

calculated by,

$$KSS = TPR - FPR = \frac{a}{(a+c)} - \frac{b}{(b+d)}, \tag{2}$$

where $a$ is the number of true positives, $b$ is the number of false positives, $c$ is the number of false negatives and $d$ is the number of true negatives (Hanssen and Kuipers, 1965).

For the case studies presented, retrievals from ORAC using the mean biomass LUT are compared to the JAXA ARP, MODIS

DB and MODIS MAIAC. The AOT data from each product is shown without any downsampling and only pixel-level quality control measures applied. For the ORAC product, this is the pixel-wise quality control, whilst for the other products, only failed retrievals are not shown. For all products, the minimum AOT is set as 0.2 to ensure that only significantly high values of AOT are shown as this study is focused on the analysis of thick plumes.

To ensure temporal matching, the AHI products that include the MODIS granule data are compared to each other, i.e. the

MODIS granule is within 10 minutes of the start of the corresponding AHI scene. The AOT data for each product is plotted over true colour RGBs generated from AHI 5 km regridded radiances, which is from the matching AHI timestamp. Within each plotted region, a 2° × 2° area is highlighted and the distribution of AOT for each product is shown in violin plots. The region is chosen to focus on the distribution of high AOT pixels and ignore erroneously retrieved cloud or background AOT that is not the focus of this study.





## 3.4 Quality control for retrievals

To maximise the number of available pixels, only failed retrievals from the JAXA ARP, MODIS, VIIRS and SLSTR products are omitted and quality control flags are ignored to ensure that high AOT pixels are included. This is done to test the accuracy and skill of the retrieval products at higher values of AOT than may be considered acceptable for operational purposes.

For ORAC retrievals, all pixels used in the collocation are included unless they are declared as cloud by the cloud mask described in Robbins et al. (2022) or the cost is greater than 5. The use of a cost threshold is important as cost is a measure of how well the retrieval describes the aerosol type in the pixel. High cost suggests that the retrieved state resulted in a poor fit between the forward model and the measurements, i.e. the pixel does not correspond to the aerosol optical properties described in the LUT or may be misclassified cloud, clear air or some other aerosol species. Applying this mask ensures a more consistent comparison between the operational retrieval products and the ORAC retrievals. The cloud mask developed in Robbins et al. (2022) was specifically developed to improve differentiation between clouds and optically thick aerosol plumes, but likely leads to some cloud contamination in AHI retrievals, which can be seen in case studies. This is considered acceptable for research purposes, where individual case studies can be checked and AERONET data can help to indicate if cloud is present in collocated pixels, depending on the application a conservative cloud mask could be used.

## 4 Results and Discussion

### 4.1 Sensitivity Analysis

For the study period, between 00:00 UTC on 01 December 2019 (inclusive) to 00:00 UTC on 03 January 2020 (exclusive), there were 1115 successful ORAC retrievals with AERONET data available. Figure 4, shows the distribution of retrieved ORAC AOT values versus AERONET AOT values. The figure indicates that all three LUTs produce a good agreement with AERONET when a linear fit is applied, with correlation of 0.758, 0.741 and 0.755 for decreased biomass, mean biomass and increased biomass respectively. The vast majority of the collocated data are clustered in the low AOT region (AOT ≤ 2) with a similar distribution across all the LUTs, whereas the distribution of high AOT values (AOT > 2) is more widely spread. This suggests that under less extreme circumstances, i.e. AOT ≤ 2, the sensitivity of the ORAC retrieval to the assumed optical properties of the biomass plume is relatively small.

However, for larger values of AOT (AOT > 2), the ORAC AOT values begin to diverge from AERONET for the decreased biomass and increased biomass LUTs. As can be seen in Fig. 4b, the mean bias for mean biomass is smallest of the three LUTs. This suggests that, under more extreme circumstances, such as the smoke plumes seen during the 2019-20 Australian bushfires, the sensitivity of the retrieval to assumed optical properties becomes much more important, which can lead to systematic under- or overestimation of AOT if the aerosol type is considered too absorbing (increased biomass) or less absorbing (decreased biomass).

Figure 5 shows that the ORAC mean biomass retrieval compared to the JAXA ARP using the strict quality controls set out in Sect. 3.2 (with the exception of the cutoff for AOT < 0.2), provides improved temporal coverage. In addition, ORAC retrieves



AOT values > 5. In this high AOT case the JAXA ARP fails due to misclassifying the aerosol as cloud. This can also be seen over the whole study period as shown in in Table 3. ORAC successfully retrieve approximately 2 times as many pixels as the JAXA ARP. When compared to the LEO satellite products, the mean values of ORAC are generally closer to AERONET values, whilst providing significantly improved coverage, successfully retrieving approximately 28 times as many pixels as the most successful LEO product (MODIS Deep Blue) with smaller values of mean bias and similar values of RMSE (see Table 2).

The LEO products have lower uncertainties when compared with the ORAC retrievals. These low uncertainties are important to note, as the AOT values and associated uncertainties generally underestimate AERONET values. In addition, Fig. 5 shows the AERONET values at higher AOT are well outside the range of uncertainty for the LEO retrieval. This is likely due to two reasons. Firstly, as can be seen in Table 2, both MODIS products are capable of retrieving high AOT plumes, with $\tau_{Max}^{A}$ values of 7.562, where AERONET data is also available. In addition, values as high as 3 can be seen in Fig. 5. Table 3 shows the skill of each satellite product flagging an aerosol event, with the truth value assigned by the presence of an AERONET retrieval. The relatively high accuracy of these products at flagging (0.778 and 0.764 for MODIS Deep Blue and MAIAC respectively) is greater than the accuracy of the ORAC retrievals (0.743, 0.744 and 0.742 for decreased biomass, mean biomass and increased biomass respectively). This indicates that cloud masking in these products is not a major issue. Instead, it would suggest that the parameterisation of the the biomass burning plumes in the MODIS retrieval products is for a type that is not absorbing enough, leading to the small values for gradient seen in Table 2. This is consistent with the results of Shi et al. (2019), where the operational MODIS DB product was found to underestimate the AOT values for the 2015 Indonesian fires. Secondly, internal limits in the algorithms can lead to systematic underestimation of the retrieved AOT. In the case of MAIAC, the maximum value of $AOT_{440\,nm}$ is set at 4 (Lyapustin and Wang, 2008). When coupled with the large values of $\alpha$ seen during biomass burning plumes, this leads to much smaller values of $\tau_{Max}^{R}$ retrieved (2.927 and 3.085 for Deep Blue and MAIAC, respectively) when compared to AERONET. In the case of the SLSTR SYN product, the low values of $\tau_{Max}^{A}$ and $\tau_{Max}^{R}$ suggest that the conservative cloud masking of the SYN algorithm prevents retrievals of extreme biomass burning plumes. The VIIRS DB product shows similar performance to the MODIS DB and MAIAC algorithms, with the same issue of systematic underestimation of AOT when compared with AERONET, suggesting the same mischaracterisation of the BB plumes is present in the VIIRS DB algorithm. It should be noted that the VIIRS DB product has the highest accuracy of all products when compared to AERONET (see Table 3), but still appears to suffer from the upper AOT limit problem.

The ORAC retrieval results indicate that the mean biomass LUT corresponds best with the biomass burning plumes throughout this study.

## 4.2   Case Studies over Australia

Three case studies from the 2019-20 bushfire season in Australia have been selected to demonstrate the performance of satellite measurements of AOT during extreme biomass plume events. The three case studies show BB plumes of varying optical thickness (AOT > 2) over a range of surface types. Regions are selected from within the case study areas to highlight where the use of the BB LUT is appropriate and where MODIS DB and MAIAC products are also available.



### 4.2.1 Moderately Thick AOT over Vegetation

The first case study over Melbourne and the SE coast of Australia from 13 January 2020 03:50 UTC shows a moderately optically thick ($2 \leq \text{AOT} \leq 5$) smoke plume (see Fig. 6). Overall, the cloud mask for the ORAC retrieval captures the optically thick regions of the smoke plume that the JAXA ARP misses (see Fig. 6c and d). In addition, the ORAC retrieval resulted in

significantly higher values of AOT when compared to the MODIS products.

Within the section of the plume shown in the $2° \times 2°$ box, Fig. 6b shows that the less optically thick region over vegetation is retrieved with reasonable agreement between all 4 products, with mean values for AOT of around 1.5 for ORAC, DB and MAIAC, whilst the JAXA ARP gives a slightly lower value. The distribution of AOT values seen in Fig. 6b suggests that, under conditions that would normally be considered extreme smoke plumes by Zhuravleva et al. (2017) and Markowicz et al. (2016),

all products perform similarly and would be suitable for use in further studies.

### 4.2.2 Moderately Thick AOT over Semi-Arid Land

In this case study, a moderately high AOT smoke plume from the 2019-20 bushfire season at 00:30 20 December 2019 UTC over the east of Australia is presented. Figure 7 shows that all of the satellite products retrieve significant portions of the biomass burning plume towards the centre of the scene. The distribution of the AOTs within the $2° \times 2°$ box for the ORAC,

Deep Blue and MAIAC products generally agree, with mean values between 2 and 3 (see Fig. 7b). However, for the JAXA ARP product shown in Fig. 7d, the most optically thick areas are not retrieved, whilst the AOT distribution (see Fig. 7b) is more widely spread compared to the other products. The peak values of this distribution are located around the edge of the more optically thick region of the plume. This, in combination with the large value for gradient in Table 2, suggests that the optical properties for biomass burning are set for a more absorbing aerosol type in JAXA ARP. This is likely to be the category

2 fine aerosol type described in Omar (2005) that is used in the JAXA ARP algorithm as described by Yoshida et al. (2018), which is not derived from data over Australia. In addition, the more optically thick plumes towards the bottom of this scene are not retrieved by the JAXA ARP, whilst the other products are capable of retrieving the AOT values. Inspection of the QA flag for the JAXA ARP indicate that the plume was correctly classified but retrievals failed. Figure 7b shows that ORAC and MODIS MAIAC have similar dsitributions, with two peaks above and below AOT values of 2, whilst the MODIS Deep Blue

product has a single, wide distribution peaking towards an AOT value of 3. The single peak is likely due to the lower spatial resolution of this product.

### 4.2.3 Extreme AOT

The extreme biomass burning plume (AOT values » 2) presented in Fig. 8 at 03:30 01 January 2020 UTC is taken during the peak of the 2019-20 Australian bushfires. Figure 8a shows that the plume is extremely optically thick and covers a large

area along the SE coast, with the most optically thick regions of the plume to the SE of the Great Dividing Range. Figure 8c shows that ORAC retrieves AOT values greater than 15, although with high uncertainty, and a mean value of 10, which are significantly higher than those seen in events classified as extreme by previous studies, where the highest values of AOT



from satellite instruments were approximately 5 (van Donkelaar et al., 2011; Markowicz et al., 2016; Petrenko et al., 2017; Zhuravleva et al., 2017; Shi et al., 2019; Mukai et al., 2021).

Both visual inspection and the cloud mask developed in Robbins et al. (2022) suggest that this plume is not cloud contaminated, but it can be seen in Fig. 8d that the JAXA ARP cloud mask misclassifies most of the BB plumes, which prevented any retrieval from being carried out in the most extreme regions of the plume. The MODIS products have not been cloud-cleared in these optically thick regions, however Fig. 8b shows that the maximum retrieved value is approximately 5, significantly lower than those found by ORAC or observed by AERONET. Of particular note is the distribution of the MODIS MAIAC AOT

values, which shows a narrow distribution of AOT around the peak value, suggesting that the MAIAC algorithm is capable of retrieving values closer to those seen by ORAC, but is constrained by the internal maximum AOT threshold of the algorithm. This is apparent in Fig. 8f, where it can be seen that the retrieved AOT values from MAIAC are almost completely homogeneous across the whole of the plume, despite significant differences observed in the true colour RGB from AHI in Fig. 8a and the ORAC retrieval in Fig. 8c. A similar issue for the Deep Blue retrieval can been seen in Fig. 8b and e, where the distribution

of AOT values is relatively narrow and heavily skewed towards the maximum value, with a notable cutoff at that maximum.

The gaps in the JAXA ARP data and underestimation of AOT in the MODIS products seen in this case study suggests that there has been a systematic underestimation of AOT values during the 2019-20 Australian bushfire season, as plumes of this magnitude were present for many days along the SE coast of Australia. This would have led to systematic underestimation in climate data sets and the estimation of radiative impacts. In addition, the lack of high quality AOT data for the full bushfire pe-

riod will have consequences for air quality (AQ) modelling, which often ingests satellite AOT products to assess concentrations of particulate matter on the ground (Inness et al., 2019).

## 5   Conclusions

In this study an aerosol property retrieval algorithm targeted at biomass burning plumes measured by the AHI instrument onboard Himawari-8 has been presented. It uses the cloud approach from the ORAC algorithm with BB LUTs developed for the

2019-20 Australian bushfire season and the NN cloud mask developed in Robbins et al. (2022). Three LUTs were developed utilising refractive index information for the BB plumes from the Tumbarumba AERONET site: decreased biomass, mean biomass and increased biomass. A sensitivity study compared available AERONET AOT values at 550 nm to ORAC retrievals using these 3 LUTs for the period between 00:00 UTC on 01 December 2019 (inclusive) and 00:00 UTC on 03 January 2020 (exclusive), showing that 1115 retrievals were successfully collocated. Linear fits of the LUT data with AERONET

showed coefficient of determination values of 0.758, 0.741 and 0.755 for decreased biomass, mean biomass and increased biomass respectively, but the mean bias of the mean biomass retrieval is smallest, suggesting that this LUT best described the optical properties of the BB plumes from the 2019-20 bushfire season. The similar clustering of low AOT values for all the LUTS but divergence of ORAC values at higher optical depths (AOT $\gg$ 2) suggests the sensitivity to assumed optical properties becomes significant during these extreme events. The ORAC retrievals were compared against JAXA ARP, MODIS

DB, MODIS MAIAC, SLSTR SYN and VIIRS DB products for the study period. The ORAC retrievals had the smallest



mean bias (0.269 for mean biomass) compared to the other satellite products, whilst retrieving higher values of AOT. This underestimation of AOT corresponds with the findings of Shi et al. (2019) and suggests that even where AOT data has been available from LEO satellite products, the optical thickness of the BB plumes have been underestimated and therefore the full impacts of the 2019-20 bushfires have likely not been studied in full.

Case studies of optically thick BB plumes from the SE coast of Australia during the 2019-20 bushfire season showed that, under less extreme conditions, the ORAC, JAXA ARP, MODIS DB and MODIS MAIAC products generally agree on AOT values for BB plumes, with the minor exception of regions of higher AOT values failing to be retrieved in the JAXA ARP. In the case of a more extreme BB plume, the ORAC retrieval found AOT values > 15, although with high uncertainty, which are significantly higher than those seen in events classified as extreme by previous studies (van Donkelaar et al., 2011; Markowicz

et al., 2016; Petrenko et al., 2017; Zhuravleva et al., 2017; Shi et al., 2019; Mukai et al., 2021). The corresponding JAXA ARP mis-classifies the more optically thick regions as cloud, whilst the MODIS products reach the upper limit of the retrieval algorithms. This suggests that there has been systematic underestimation of AOT values throughout the 2019-20 bushfire season and implies that similar scale fires which produced BB plumes with AOT > 5 have not been studied in full. As this data can be used for air quality modelling, the lack of satellite AOT data may impact health studies and AQ forecasts for people in

regions impacted by extreme plumes. The omission of extreme plumes will affect long-term climatological studies, particularly those investigating the impacts of BB on the atmosphere and weather. However, it should be noted that this study covers the development of a scientific product, not an operational one, and further work will be needed to balance the cloud approach for extreme plumes with the aerosol approach for typical conditions in an operational setting.

Throughout this study, there has been a lack of ground-based instrumentation to validate AOT retrievals, as well as a lack

of active instrument data that is collocated with cloud-free BB plumes. This makes validating satellite products over Australia challenging. Over the period covered in this study, only the Tumbarumba AERONET site was active in the vicinity of the more extreme BB plumes. In addition, there are no publicly available ground-based lidar sites in this region. Overall, the lack of ground-based instrumentation across Australia, and particularly in more populated regions, poses a serious challenge to any study of aerosols and satellite AOT retrievals in the region. Therefore, an increase in ground-based sites would provide the

opportunity to comprehensively validate the satellite products during the next extreme bushfire event, leading to higher quality satellite products for Australia and improved forecasts.

## Appendix A: Himawari-8 Band 3 versus MODIS Green Bands

The ORAC algorithm makes use of the MODIS BRDF product to estimate the surface reflectance of a clear sky scene for use in the aerosol and cloud retrievals. This includes reflectances across the visible spectrum, such as the green vegetation bands

in bands 5-7 (see Fig. A1). However, the equivalent of the green band in Himawari-8 AHI is centred at 0.51 μm, which is not centered on the reflectance peak of healthy vegetation of 0.55 μm (Knipling, 1970) and leads to vegetated areas appearing dark in true colour imagery (Miller et al., 2016). With respect to ORAC retrievals utilising the MODIS BRDF, this leads to a significant difference between expected reflectances in the green band, as it can be seen that there is very little overlap between



the AHI green band and the MODIS green bands in Fig. A1. In addition, under the conditions seen during the 2019-20 bushfire

season, large swaths of land were burnt, changing the colour of the surface significantly in the MODIS BRDF, but less so in the AHI green band. Therefore, although the cloud approach of ORAC is less sensitive to the characterisation of the surface, to ensure that this difference does not cause issues in the retrieval due to the significant uncertainty associated with the different characterisations of vegetated surface reflectance, the Himawari-8 AHI band 2 is not included in the retrievals.

## Appendix B: VIIRS and SLSTR Case Studies

The VIIRS and SLSTR AOT products are not collocated within the 10 minutes of MODIS and AHI products, so aren't included in the case studies section. However, the closest spatially and temporally matching products are presented in Fig. B1 for each case study. Figures B1a and c show that the VIIRS product has similar performance to the MODIS products during less optically thick BB plumes, but significantly underestimates AOT for the more optically thick plume seen in Fig. B1e. Figures B1b, d and f all demonstrate that the SLSTR SYN product has no data available for large areas within BB plumes and generally has

significantly lower values of AOT when compared to the other operational AOT products, suggesting that AOT values will be even more severely underestimated in climatologies derived from this data than climatologies derived from other operational AOT products.

*Code and data availability.* The ORAC code is open source and available from GitHub under the GNU General Public License (GPL), Version 3 (https://github.com/ORAC-CC/orac, last access: 04 August 2023; Thomas et al. (2009); Poulsen et al. (2012); McGarragh et al.

(2018)). The ORAC data for the BB plumes, including the time-series and case studies data, are available on request from the corresponding author. AERONET data is available from the AERONET site (https://aeronet.gsfc.nasa.gov/cgi-bin/data_display_aod_v3, last access: 04 August 2023; Holben et al. (1998); Giles et al. (2019)). The AHI L1b data used in the ORAC retrievals is available on NCI Gadi (Bureau Of Meteorology, 2022). The JAXA ARP is available from the P-TREE system (https://www.eorc.jaxa.jp/ptree/, last access: 04 August 2023; Yoshida et al. (2018)). The MODIS, SLSTR and VIIRS products are available from the NASA EarthSearch system (https://search.earthdata.

nasa.gov/search, last access 04 August 2023; North and Henkel (2010); Lyapustin et al. (2018); Sayer et al. (2018); Hsu et al. (2019); Sayer et al. (2019)).

*Author contributions.* DJVR led the study, carried out the analysis and wrote the manuscript. CAP assisted in the analysis, provided scientific advice and assisted in the writing of the manuscript. STS assisted in the analysis of results, provided scientific advice during the study and helped to edit and review the manuscript. SRP assisted in the development of code and assisted in the analysis. ATP assisted in the

development of code and helped to edit and review the manuscript. RGG created the new Aerosol LUTs and assisted in the analysis. ACP assisted in the development of code.



*Competing interests.* The authors declare that they have no conflict of interest.

*Acknowledgements.* This research/project was undertaken with the assistance of resources and services from the National Computational Infrastructure (NCI), which is supported by the Australian Government. This study was partly funded through NERC's support of the

National Centre for Earth Observation, award number NE/R016518/1. We thank Ian Lau and David Barker for their effort in establishing and maintaining the Tumbarumba site.



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







**Figure 1.** Digital elevation model (DEM) derived from Geoscience Australia data (Hutchinson et al., 2008) of the SE coast of Australia with the locations of major cities (stars) and the Tumbarumba AERONET site (red circle). The DEM shows the Great Dividing Range dominating the topography of the SE coast, extending all the way from Victoria in the south of mainland Australia to Queensland in the north. The inset map shows the wider context of the region, with the region of interest highlighted by the black square, in relation to the Australian region.

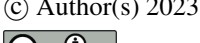



**Table 1.** Description of Himawari-8 Advanced Himawari Imager bands and their associated purposes (Bessho et al., 2016; Zhang et al., 2018; Robbins et al., 2022).

| Band Number | Central Wavelength [μm] | Spatial resolution at nadir [km] | Purpose |
|:---:|:---:|:---:|:---:|
| 1 | 0.47 | 1 | Aerosols |
| 2 | 0.51 | 1 | Composite imaging |
| 3 | 0.64 | 0.5 | Vegetation; aerosol over water |
| 4 | 0.86 | 1 | Cirrus clouds; vegetation |
| 5 | 1.6 | 2 | Phase; particle size; snow |
| 6 | 2.3 | 2 | Land; phase; particle size; snow |
| 7 | 3.9 | 2 | Clouds; night time fog |
| 8 | 6.2 | 2 | High altitude water vapour |
| 9 | 6.9 | 2 | Mid altitude water vapour |
| 10 | 7.3 | 2 | Low altitude water vapour |
| 11 | 8.6 | 2 | Total atmospheric water; cloud phase; dust |
| 12 | 9.6 | 2 | Ozone |
| 13 | 10.4 | 2 | Surface temperature; clouds; atmospheric window |
| 14 | 11.2 | 2 | Clouds; atmospheric window |
| 15 | 12.4 | 2 | Total water; ash; atmospheric window |
| 16 | 13.3 | 2 | Air temperature; cloud height |





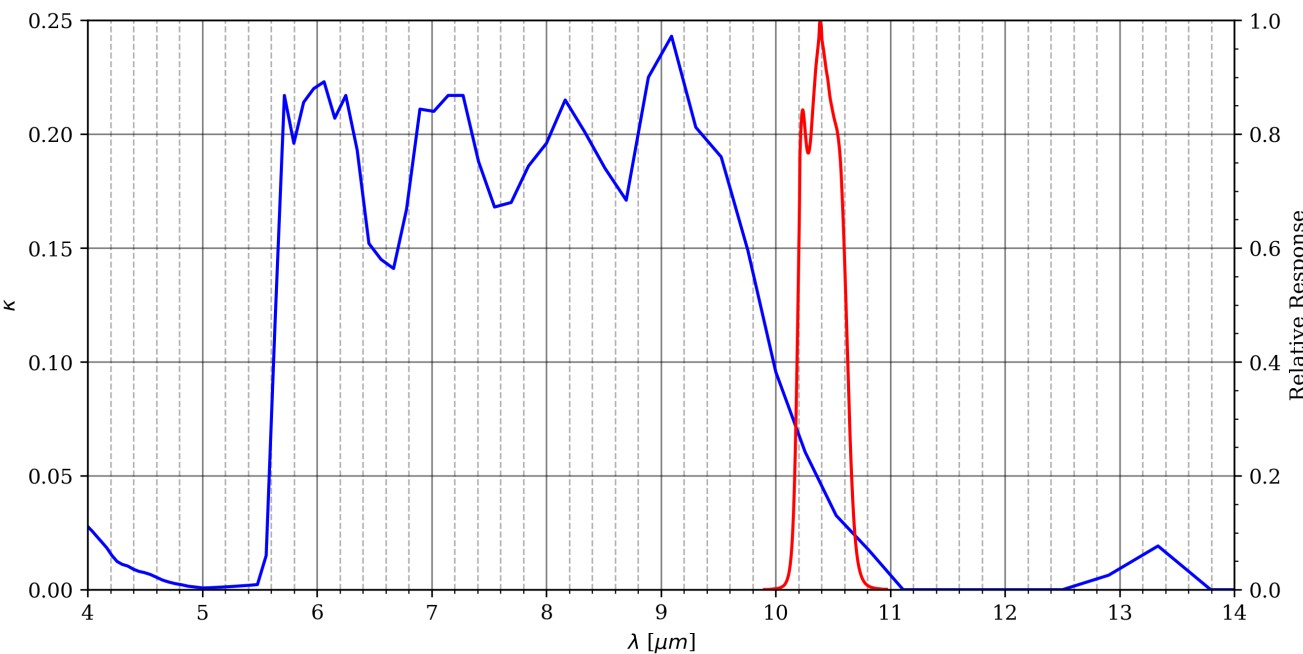

**Figure 2.** The spectral response for band 13 in Himawari-8 AHI (red, https://www.data.jma.go.jp/mscweb/en/himawari89/space_segment/ srf_201309/AHI-08_SpectralResponsivity.zip; last access 21 March 2023) overlaid onto the imaginary component of the refractive index for biomass burning described in Sutherland and Khanna (1991) (blue).



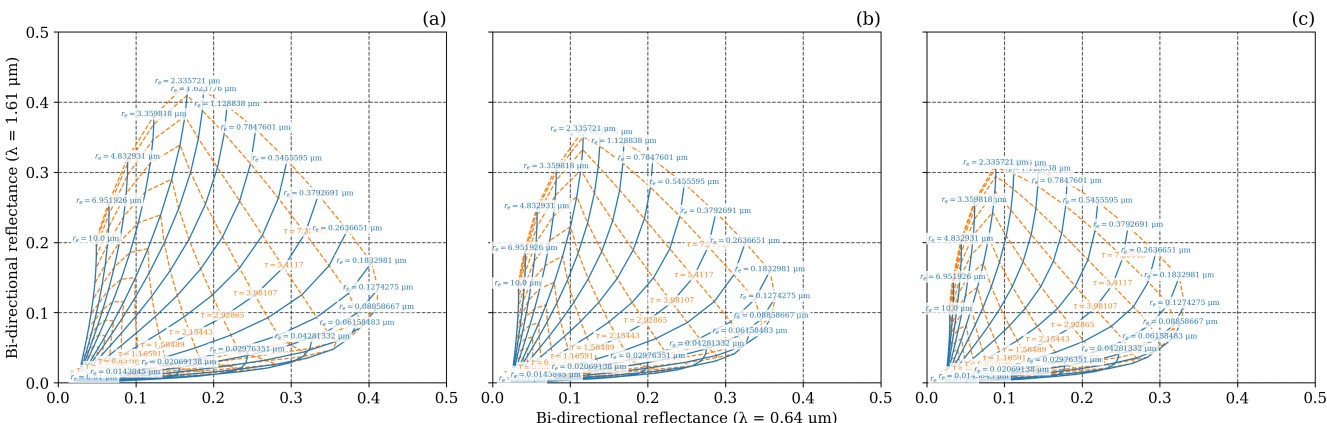

**Figure 3.** Nakajima-King plots for the (a) decreased biomass, (b) mean biomass and (c) increased biomass LUTs at a solar zenith angle of 20°, satellite zenith angle of 40° and realtive azimuth angle of 54°. These values are chosen as they are approximately the viewing angle of AHI to the Tumbarumba site with a solar zenith angle at approximately mid-day during the study period. The optical depth (orange) and effective radius (blue) are plotted for AHI band 3 (0.64 μm) on the x-axis and AHI band 5 (1.61 μm) on the y-axis.



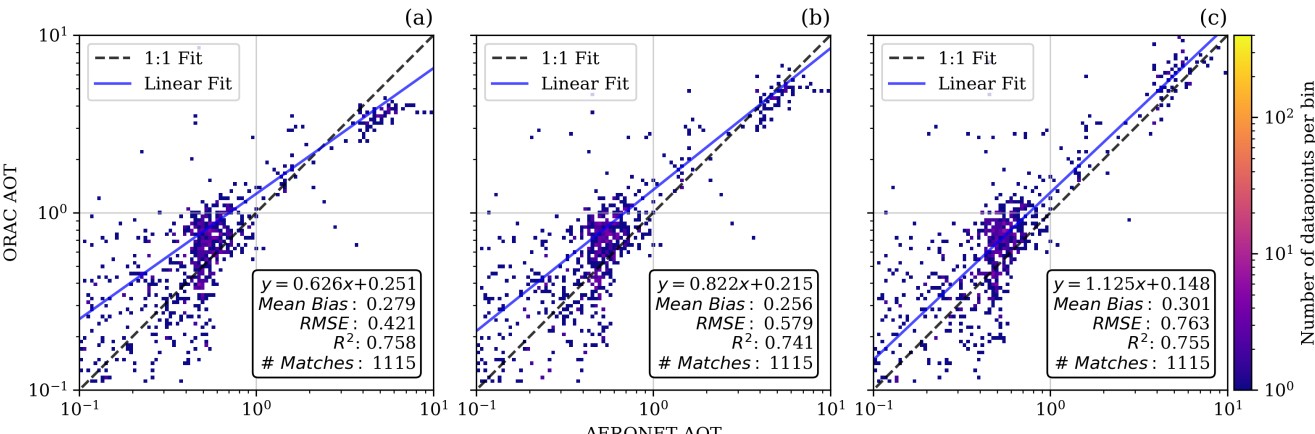

**Figure 4.** Heatmaps of the distribution of collocated ORAC AOT values versus AERONET AOT values at 550 nm for (a) decreased biomass, (b) mean biomass and (c) increased biomass LUTs, along with the linear fits (blue line), 1:1 fit (dashed black line) and associated statistics for each LUT.





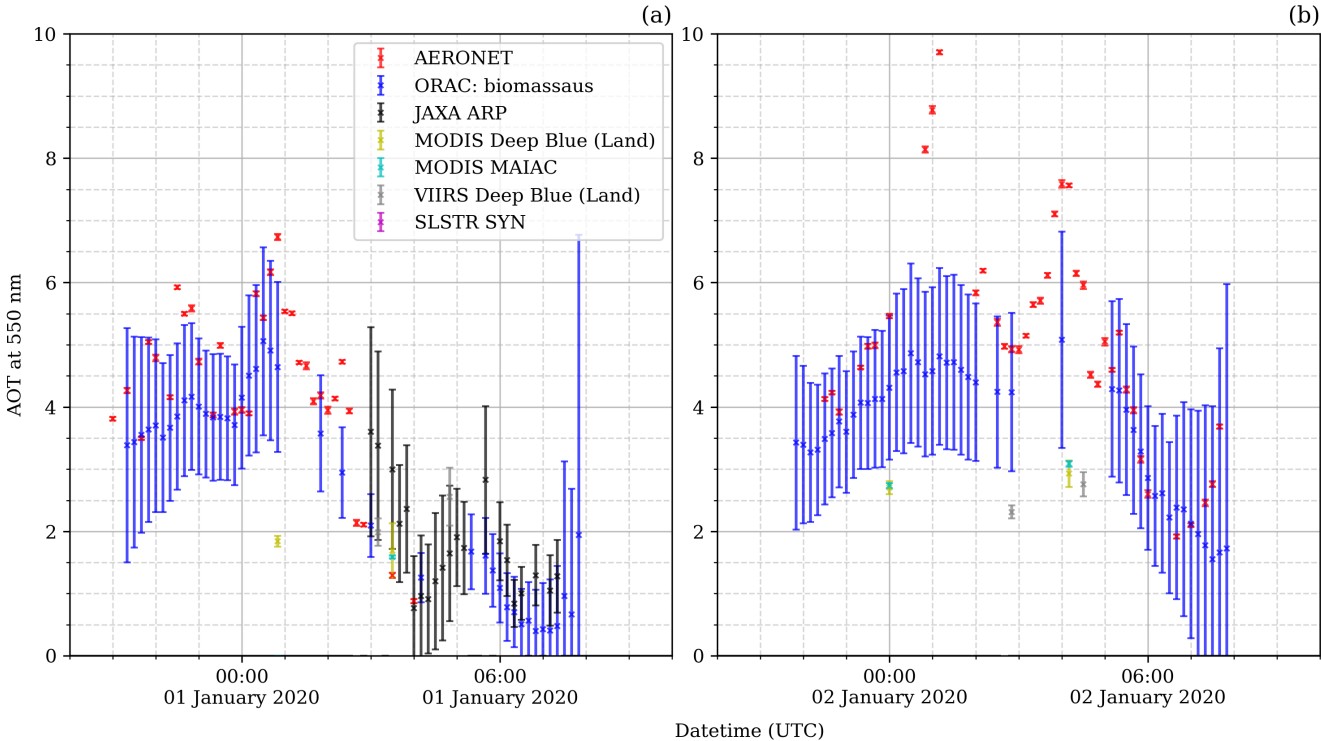

**Figure 5.** Time series for (a) 31 December 2019 - 01 January 2020 and (b) 01 January 2020 - 02 January 2020 of the retrieval products used in the sensitivity study during the peak values seen in the AERONET data. For the ORAC retrievals, only the mean biomass values are shown to prevent significant clutter in the figure. Note that there are no SLSTR SYN data plotted as there are no successful retrievals for this product in this time period.





**Table 2.** Resulting statistics of collocations between the retrieval products and AERONET where both retrievals are successful for the whole study period. The number of collocations where both the retrieval product and AERONET are successful are shown in the first column. The second and third column show the maximum retrieved AOT value in the study period where both retrievals are successful for AERONET and the retrieval product respectively. The fourth and fifth columns show the gradient and offset for the linear fit between the retrieval product and AERONET. The mean bias of the collocated data and the root mean squared error (RMSE) are shown in the sixth and seventh column respectively. Finally, the Pearson correlation coefficient is shown in the final column.

| Retrieval Product | # Matches | $\tau_{Max}^{A}$ | $\tau_{Max}^{R}$ | Gradient | Offset | $R^2$ | Mean Bias | RMSE | $r$ |
|---|---|---|---|---|---|---|---|---|---|
| ORAC: decreased biomass | 822 | 9.701 | 3.823 | 0.581 | 0.309 | 0.813 | 0.269 | 0.303 | 0.902 |
| ORAC: mean biomass | 826 | 9.701 | 5.080 | 0.728 | 0.296 | 0.841 | 0.269 | 0.355 | 0.917 |
| ORAC: increased biomass | 820 | 9.701 | 7.552 | 0.961 | 0.267 | 0.860 | 0.291 | 0.439 | 0.927 |
| JAXA ARP | 378 | 1.675 | 3.632 | 1.567 | 0.013 | 0.730 | 0.298 | 0.237 | 0.855 |
| MODIS Deep Blue (Land) | 30 | 7.562 | 2.927 | 0.346 | 0.285 | 0.698 | 0.578 | 0.432 | 0.835 |
| MODIS MAIAC | 21 | 7.562 | 3.085 | 0.433 | 0.155 | 0.908 | 0.420 | 0.258 | 0.953 |
| SLSTR SYN | 14 | 0.666 | 1.136 | 1.470 | 0.151 | 0.925 | 0.293 | 0.108 | 0.962 |
| VIIRS Deep Blue (Land) | 25 | 5.957 | 2.755 | 0.449 | 0.186 | 0.860 | 0.355 | 0.259 | 0.927 |





**Table 3.** Comparison between satellite versus AERONET retrievals showing the skill of each satellite product at flagging an aerosol event successfully. The truth label is set by AERONET, i.e. if AERONET provides an AOT value, the label is 1 (BB present), but if AERONET does not provide an AOT value, the label is 0 (masked as cloud or too optically thick to retrieve information). It should be noted that this may introduce some bias where high AOT values in the retrieval products are classified as false positives. The first column provides the total number of pixels collocated between the satellite product and AERONET regardless of if either retrieval is succesful. The second column shows the number of matches, which is equivalent to the number of true positives (TP) for the satellite product assuming AERONET is truth. The third column shows the true positive rate (TPR) and the forth column shows the false positive rate (FPR). The fifth column shows the accuracy of the satellite product. The final column shows the Kuiper skill score (KSS) for the retrieval product (Hanssen and Kuipers, 1965).

| Retrieval Product | Total Pixels | # Matches | TPR | FPR | Accuracy | KSS |
|---|---|---|---|---|---|---|
| ORAC: decreased biomass | 2493 | 822 | 0.653 | 0.165 | 0.743 | 0.488 |
| ORAC: mean biomass | 2493 | 826 | 0.656 | 0.167 | 0.744 | 0.489 |
| ORAC: increased biomass | 2493 | 820 | 0.651 | 0.165 | 0.742 | 0.486 |
| JAXA ARP | 2460 | 378 | 0.304 | 0.117 | 0.591 | 0.187 |
| MODIS Deep Blue (Land) | 72 | 30 | 0.968 | 0.366 | 0.778 | 0.602 |
| MODIS MAIAC | 72 | 21 | 0.583 | 0.056 | 0.764 | 0.528 |
| SLSTR SYN | 39 | 14 | 0.667 | 0.222 | 0.718 | 0.444 |
| VIIRS Deep Blue (Land) | 48 | 25 | 0.962 | 0.318 | 0.833 | 0.643 |



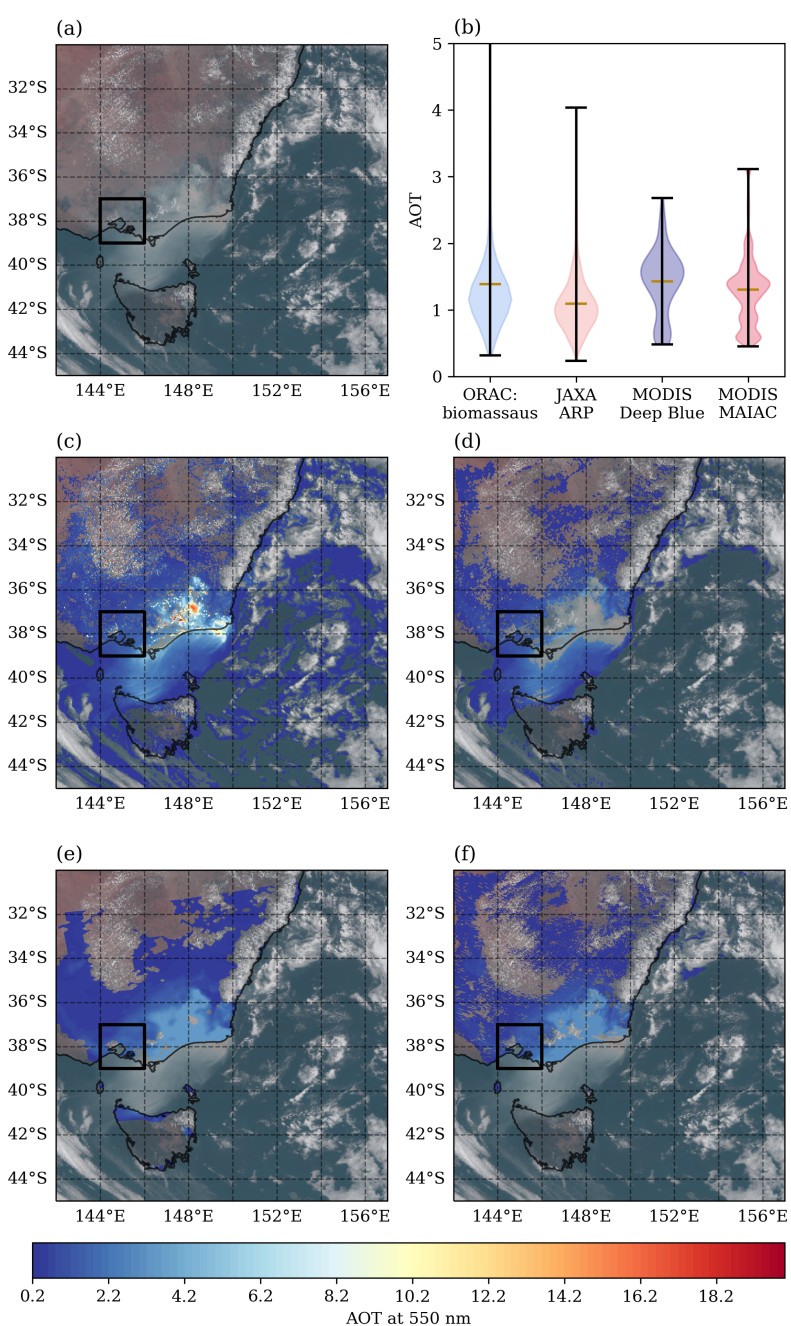

**Figure 6.** Relatively low AOT smoke over vegetation for 13 January 2020 03:50 UTC. The true colour RGB from AHI is shown in panel (a). Panel (b) shows violin plots of the AOT values within the highlighted box, indicating the frequency of AOT values, for (c) ORAC mean biomass, (d) JAXA ARP, (e) MODIS Deep Blue and (f) MODIS MAIAC AOT retrievals.





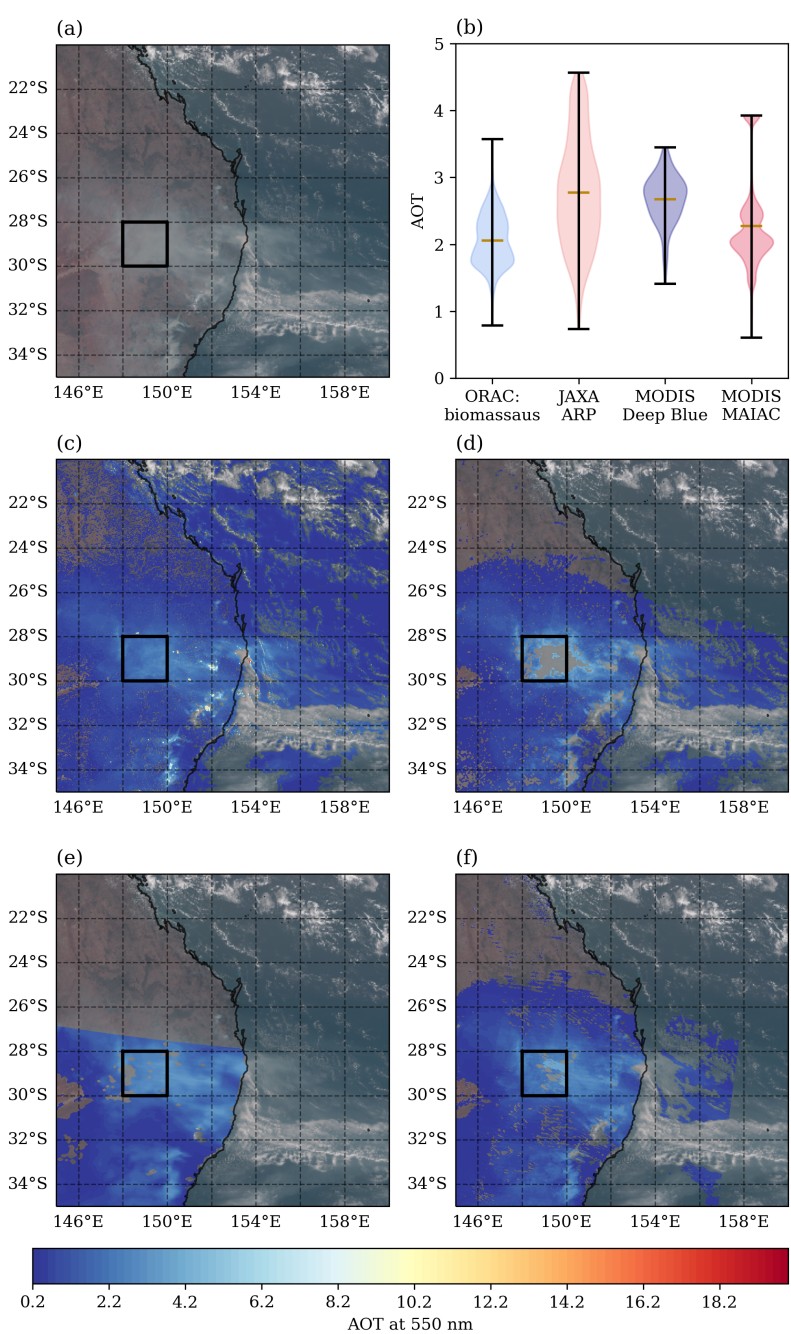

**Figure 7.** Low AOT smoke over semi-arid land for 20 December 2019 00:30 UTC. The true colour RGB from AHI is shown in panel (a). Panel (b) shows violin plots of the AOT values within the highlighted box, indicating the frequency of AOT values, for (c) ORAC mean biomass, (d) JAXA ARP, (e) MODIS Deep Blue and (f) MODIS MAIAC AOT retrievals.



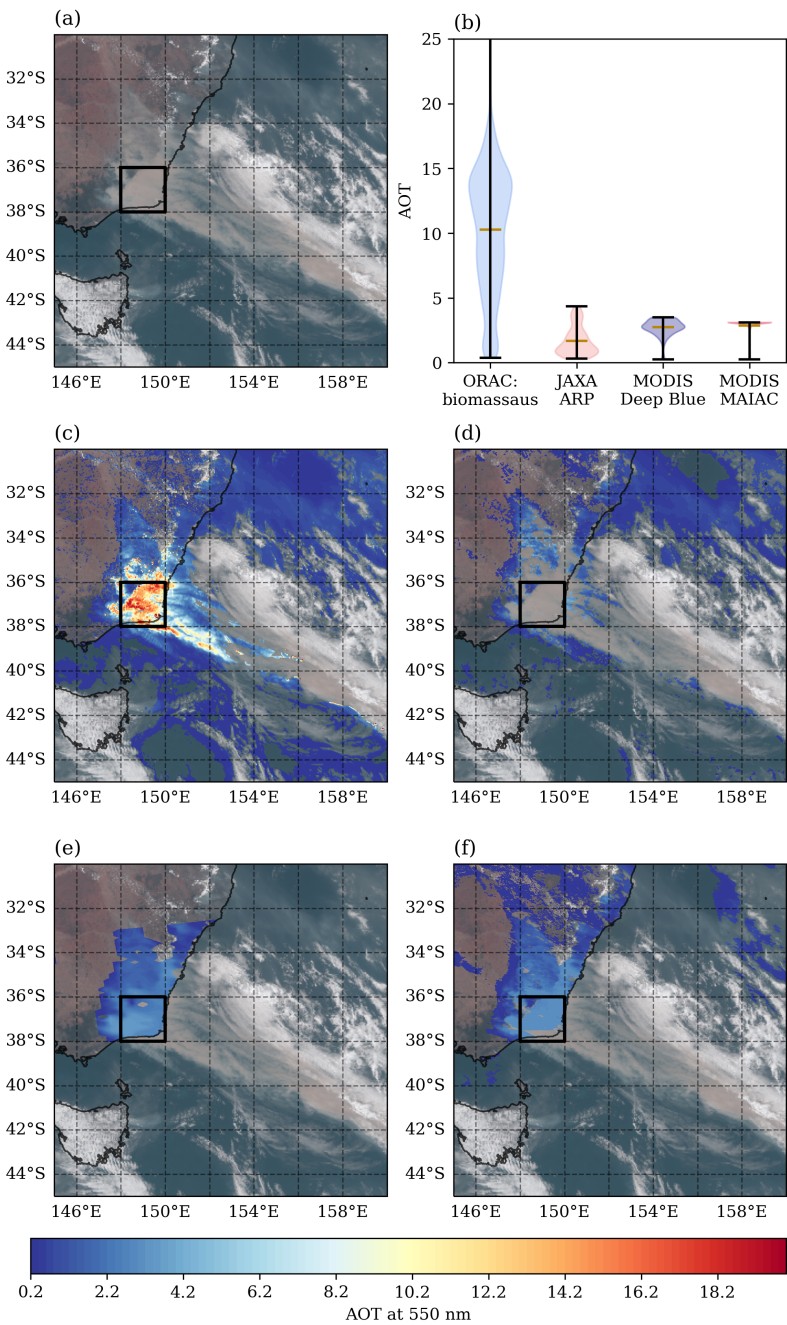

**Figure 8.** High AOT smoke for 01 January 2020 03:30 UTC. The true colour RGB from AHI is shown in panel (a). Panel (b) shows violin plots of the AOT values within the highlighted box, indicating the frequency of AOT values, for (c) ORAC mean biomass, (d) JAXA ARP, (e) MODIS Deep Blue and (f) MODIS MAIAC AOT retrievals.



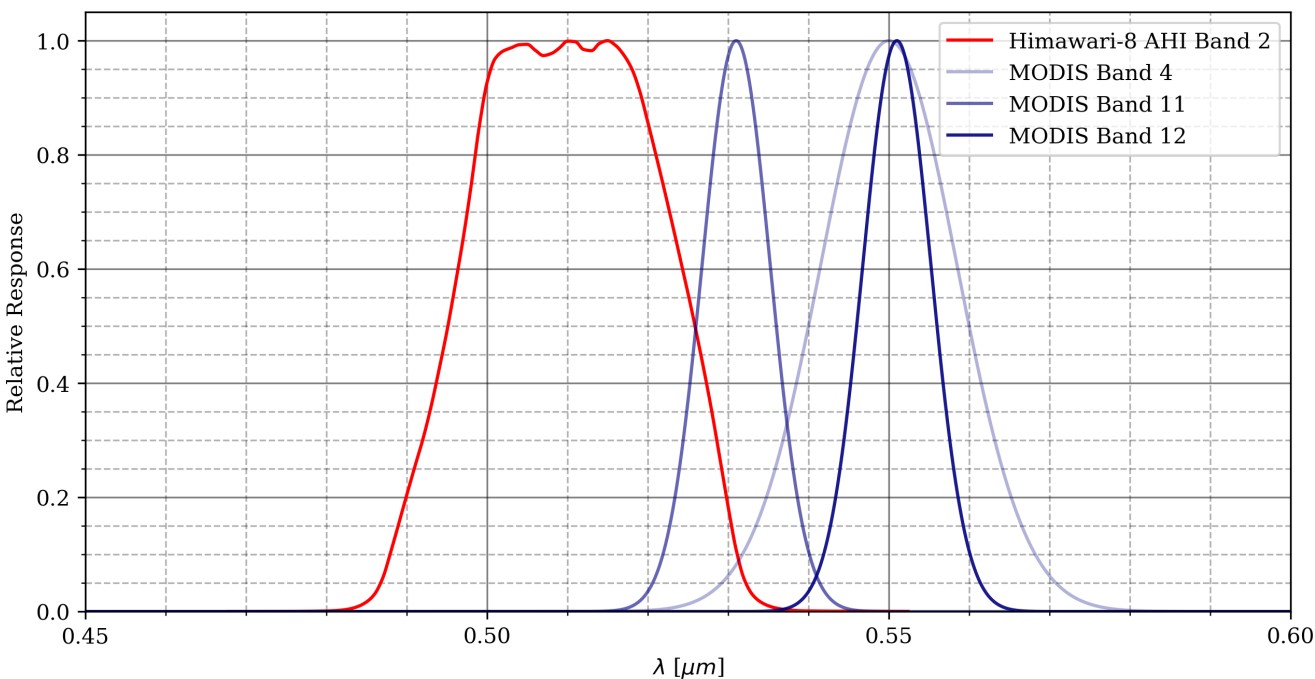

**Figure A1.** The relative spectral response for band 2 in Himawari-8 AHI (red, https://www.data.jma.go.jp/mscweb/en/himawari89/space_
segment/srf_201309/AHI-08_SpectralResponsivity.zip; last access 21 March 2023) along with the nominal response for bands 4, 11 and
12 in MODIS (blue, https://modis.gsfc.nasa.gov/about/specifications.php; last access 04 August 2023). The AHI band is considered a green
band, but doesn't significantly overlap with the green bands of the MODIS instruments.





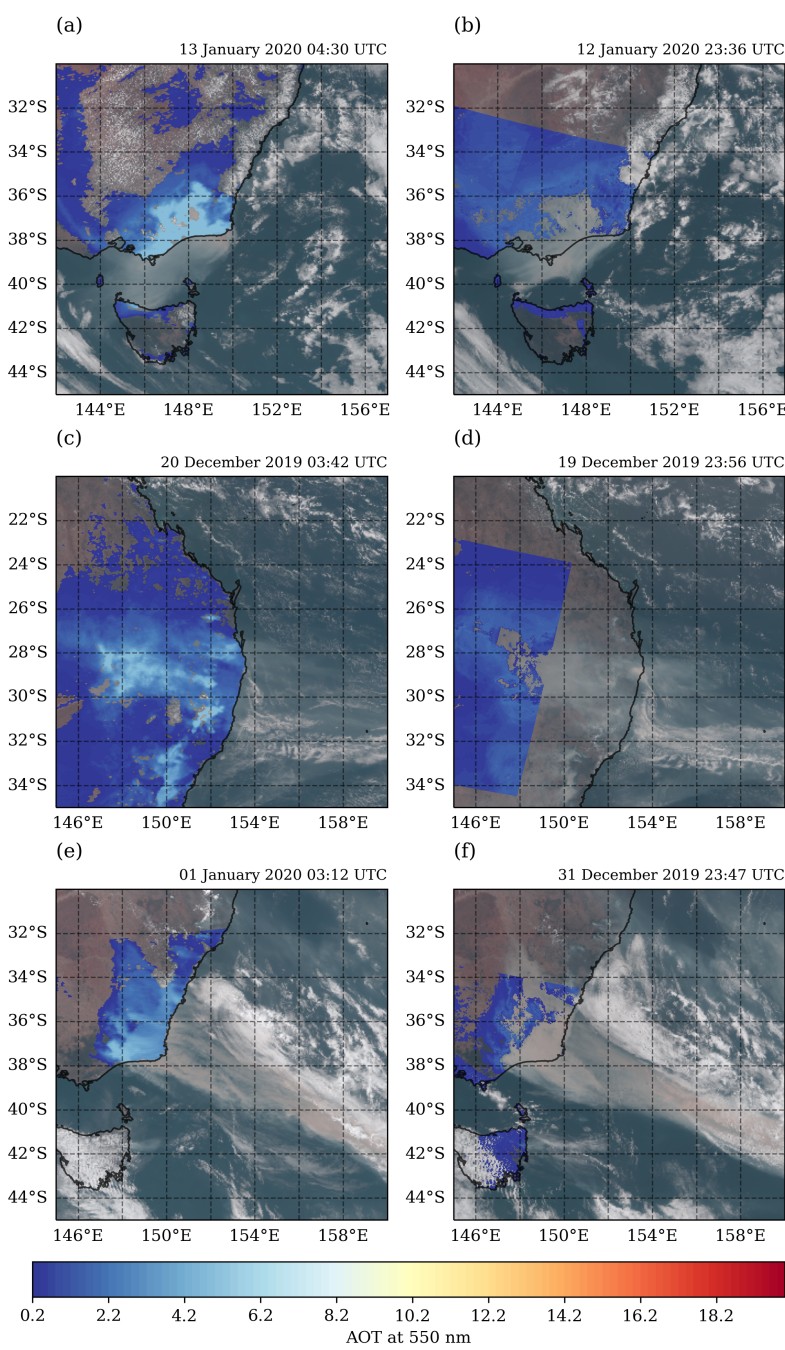

**Figure B1.** The closest matching VIIRS DB (left) and SLSTR SYN (right) products for the case studies presented in Sect. 4.2. The top row, (a) and (b), matches with the first case study, the middle row, (c) and (d) with the second case study and the bottom row, (e) and (f) with the third case study.