# Peer review of "Geostationary aerosol retrievals of extreme biomass burning plumes during the 2019-20 Australian bushfires"

_EGUsphere, 2023_

## Author Response (AR1)

**Author's Response**

The response to the reviewers is set out below, with reviewer comments in black and author's response in red, which includes the relevant changes to the manuscript. A complete list of the changes to the manuscript are set out at the end of this document.

**Reply to Reviewer 1 (Antti Lipponen)**

Thank you very much for your kind review and comments. We have addressed your concerns in the manuscript as follows (reviewer comments are in red, authors comments are in black beneath the corresponding reviewer comment):

l.12 "L1.5" please write in full "level 1.5"

We have updated "L1.5" to "level 1.5" in line 12.

l.15 "...MODIS MAIC SLSTR SYN..." comma missing

We have added a comma to line 15.

l.45 "...can accurately monitor aerosol properties in the atmosphere directly above the site location.." To be precise, especially with larger solar zenith angles, the AERONET is also sampling some nearby station locations, not just the pointwise location. You may want to modify the text accordingly, to be precise.

Thank you for this clarification, we have updated line 45 from "...can accurately monitor aerosol properties in the atmosphere directly above the site location" to "...can accurately monitor atmospheric aerosol properties in the area above the site location".

l.64 "..in recent years. such as..." extra .

This was a typo in the manuscript, the full stop should have been a comma. This has been updated in the manuscript.

l.88 Please add the wavelength for the CALIOP AOT

The CALIOP AOTs were found from the 532 nm channel. Line 88 has been updated to include this information.

l.89 Please add the wavelength for the AOT listed on this line

The AOT values were at 550 nm and this information has been added to the manuscript.

l.95 Please mention if the Zhuravleva et al. (2017) estimates were AERONET or satellite (which instrument/algorithm) estimates

The AOT values described on this line are derived from L3 MODIS daily products, which are based on the Dark Target and Deep Blue algorithms, presented in Table 5 of Zhuravleva et al. (2017). However, the authors do not clarify which algorithm the value is taken from, so we have updated lines 95 and 96 as best as we can from, "...The 2012 Western Siberian fires were found by Zhuravleva et al. (2017) to reach $AOT_{550\ nm}$ of approximately 3.54 with subsequent significant cooling effects on the surface…" to "...The 2012 Western Siberian fires were found by Zhuravleva et al. (2017) to reach $AOT_{550\ nm}$ of approximately 3.54 in MODIS level 3 daily data, with subsequent significant cooling effects on the surface…".

l.130 Please confirm/clarify what exact AERONET data you use, the Direct Sun or Aerosol Inversions data? Now, it is not clear.

We have used the Direct Sun measurements for this study. Line 134 has been updated from "... version 3 level 1.5 AOT data…" to "... version 3 level 1.5 Direct Sun AOT data…".

l.158 Please consider changing SLSTR to Sentinel-3 Synergy as the Synergy data is (mostly) based on both SLSTR and OLCI instruments' data both flying on Sentinel-3 satellites. If you decide to change, also change SLSTR throughout the manuscript to "Sentinel-3 Synergy" or similar.

Thank you for this clarification, we agree with you about the choice of name throughout the manuscript. We have updated all instances of "SLSTR SYN" to "Sentinel-3 SYN", updated the heading in line 158 to "Sentinel-3" and updated the label in Figure 5.

l.161 Please clarify if you use the SYN surface reflectance and aerosol parameter (SY_2_SYN) or SYN AOD data (SY_2_AOD) product that are different Synergy products. If you used the SY_2_SYN data product, please justify why this data product, whose primary use is not aerosol information but surface reflectance, is used instead of the dedicated aerosol data product (SY_2_AOD).

The SYN AOD data is not available for the time period we are interested in. The data in the Copernicus Data Space Ecosystem [last access: 20 December 2023] starts around 19 Feb 2020; the first file available seems to be: S3A_SY_2_AOD____20200219T053103_20200219T061449_20200220T124211_2626_055_105______LN2_O_NT_002.SEN3

We have included this information in the manuscript by updating line 161 to read "For the period of this study, the L2 AOD product is not available, but the SYN product is available,...".

l.400 Please define "NN"

Thank you, we have added the definition to line 400.

**Reply to Reviewer 2 (Marloes Penning de Vries)**

We would like to thank you very much for your kind review and comments. We have addressed your comments as follows (reviewer comments are in red, authors comments are in black beneath the corresponding reviewer comment):

First: the dependence of AOT retrieval on viewing angle. The effect of parallax on a plume of moderate altitude is mentioned, but the strongly elongated light path due to large viewing (and/or solar) angles is not discussed. This, I imagine, is a significant issue for regions at the edges of the geostationary field of view.

We agree that viewing angle is a significant issue for geostationary retrievals, as the effects of parallax and rayleigh scattering become pronounced towards the edges of the field of view and lead to large uncertainties on any retrieval. However, the ORAC algorithm takes into account the path length due to viewing angles when carrying out a retrieval. More information can be found on this in the published works discussing ORAC, e.g: Prata et al., 2022, McGaragh et al., 2018, Poulsen et al., 2012 and Thomas et al., 2009. For the purposes of this study, we aren't concerned by these effects as we focus on high AOT tropospheric plumes that occur towards the centre of AHI's field of view. However, the effects from high solar zenith angles can be seen towards the beginning and end of the day in Figure 5, where the uncertainties on ORAC retrievals begin to grow, but are a known issue with no obvious solution.

Second: the authors state that at higher AOT the retrieval depends more on the assumed aerosol parameters (particle size distribution, refractive index) than for smaller AOT. The ORAC algorithm performs very well for the Tumbarumba station, but then it makes use of the aerosol parameters derived from Tumbarumba AERONET data. How does this translate to a global algorithm? The implication appears to be that extreme AOT values can only be retrieved from satellite if the aerosol parameters are accurately known for the region (or even fire event) in question. I'd like to read the authors' view on this point. Because although they stress that their algorithm is scientific and not operational, it may be assumed that they, or others, will continue the advancement of aerosol algorithms towards routine monitoring of even extreme biomass burning cases.

We have had a similar discussion amongst ourselves on this topic which we did not believe was suitable to go into the manuscript, but we are more than happy to discuss this issue in this reply.

The results of the sensitivity analysis do suggest that accurate optical properties are needed to carry out accurate retrievals of extremely high AOT plumes. In theory, this would mean that for accurate retrievals across the globe, we would need to know the specific optical properties for each event, which is not always practical. If there were an AERONET site nearby that captures the plume, optical properties could be derived using the technique we have employed in this study. However, for the majority of events, we are unlikely to have this information. Therefore, moving towards more specific optical property models could be a more practical approach, e.g. for Australia, deriving look-up tables (LUTs) for biomass-burning (BB) events for grassland, scrubland, tropical forest and temperate forest could be more appropriate than using simple Australia-wide BB LUT. These would be based on climatologies of the specific types of events, limiting the number of LUTs needed in an operational retrieval algorithm and producing time series data that is unbiased.

However, reprocessing BB events with specific optical properties and comparing them to the data produced using the more practical approach would be an exciting area of research to assess the applicability of BB LUTs in a global context.

**Changes to the manuscript**

The following changes have been made to the manuscript:

- Upon request from Andrew Prata, Prata's affiliation has been updated to include "now at CSIRO…". This has required the addition of a new \textdagger affiliation and has required Adam Povey's \textdagger affiliation to be changed to a \textdaggerdbl.
- Line 12: "L1.5" has been updated to "level 1.5"

- Line 15: "SLSTR" has been updated to "Sentinel-3". All cases of "SLSTR" have been changed to "Sentinel-3" and there are many cases of this throughout the manuscript, including in Tables 2 and 3, Fig. 5 and the description of Fig. B1. To prevent cluttering of this document with noting every instance, the reader is directed to the tracked-changes pdf.
- Line 46: Addition of "atmospheric", removal of "atmosphere directly" and addition of "area".
- Line 89: Addition of subscript "532 nm" to "AOT".
- Line 90: Addition of subscript "550 nm" to "AOT".
- Line 97: Addition of "in MODIS level 3 daily data".
- Line 135: Addition of "Direct Sun".
- Line 162: Addition of "AOD product is not available, but the".

---

## Author Response (AR2)

**Authors' Response**

We would like to thank the editor, Marloes Penning de Vries, for their prompt review and kind comments. We have integrated responses to their requests in the manuscript as follows (editor's request in black italics, authors' response in red beneath the corresponding request):

*(1) a sentence regarding the viewing angle dependence, e.g., your statement that "the effects from high solar zenith angles can be seen towards the beginning and end of the day in Figure 5, where the uncertainties on ORAC retrievals begin to grow, but are a known issue with no obvious solution"*

We have added a short statement to lines 319-321 describing the increasing uncertainty associated with the decreased signal-to-noise ratio at large viewing and solar angles.

*(2) one or two sentences on the global applicability of your approach, which could be a condensed version of your reply to my second reviewer comment.*

We have added a short paragraph explaining the global applicability of the techniques described in this manuscript and the feasibility of transfer this to an operational product to lines 441-447.